# Chemometrics-aided surface-enhanced Raman spectrometric detection and quantification of GH and TE hormones in blood

Annah M. Ondieki[1]*, Zephania Birech[1], Kenneth A. Kaduki[1], Peter W. Mwangi[2], Moses Juma[1,3], Boniface M. Chege[2,4]

**1** Laser Physics and Spectroscopy Research Group, Department of Physics, University of Nairobi, Nairobi, Kenya, **2** Department of Medical Physiology, University of Nairobi, Nairobi, Kenya, **3** UNESCO-UNISA Africa Chair in Nanoscience/Nanotechnology, College of Graduate Studies, University of South Africa (UNISA), Pretoria, South Africa, **4** School of Health Sciences, Dedan Kimathi University of Technology, Dedan Kimathi, Kenya

* moraa94annah@gmail.com

## Abstract

Growth hormone (GH) and testosterone (TE) levels in blood are crucial indicators of human health and performance in clean sports. Deviations from normal levels can signal serious health issues, such as fertility problems, cancer, or pituitary tumors. Existing detection methods for these hormones are often costly, time-consuming, and lack portability. In this study, we explored the potential of Surface-Enhanced Raman Spectroscopy (SERS) in distinguishing blood samples from Sprague Dawley (SD) rats injected with exogenous GH, TE and both hormones from those not injected. Then, used artificial neural network (ANN) models trained, and validated in predicting levels of these hormones in blood. Blood samples from SD rats injected with GH, TE, both hormones, and non-injected rats were analyzed using the SERS method upon 785 nm laser excitation. The recorded Raman spectra from blood of GH and TE injected and non-injected rats displayed hormone-specific band intensity variations. Additionally, Principal Component Analysis (PCA) showed temporal changes in band intensities post-injection, suggesting hormone-induced biochemical alterations. In particular, Raman bands centered around 1378 cm⁻¹ for all groups, 658 cm⁻¹ for GH, and 798 cm⁻¹ for GH and TE displayed significant intensity variations. The ANN models, trained using PCA scores from blood samples with varied hormone concentrations, achieved high predictive accuracy with coefficients of determination ($R^2 > 87.71\%$) and low root mean square error (RMSE < 0.6436). Elevated hormone levels were initially observed in injected rats, gradually declining over time, with results aligning closely to those obtained via ELISA kits. This work showed that the SERS method can provide rapid (~2 minutes), hormone-independent detection with minimal sample preparation. This approach demonstrated the SERS method's potential for rapid, reliable hormone detection and with customized calibration may be applied in sports doping control, clinical diagnostics, and broader biomedical research.

**Data availability statement:** The data is available in Github: https://github.com/Moraa1714/Matlab-codes-and-data-on-hormone-quantification.

**Funding:** Swedish International Development Cooperation Agency (SIDA) through the International Science Programme (ISP), Uppsala University, KEN:04, Prof. Kenneth A. Kaduki.

**Competing interests:** There are no financial interests or personal relationships that could have appeared to influence the work reported in this paper and potential competing interests don't exist.

## 1. Introduction

Growth hormone (GH) and Testosterone (TE) hormones are essential for regulating numerous physiological functions (i.e., muscle growth, development and strength, bone density and sexual health) in humans [1,2]. Imbalance of these hormones can suggest endocrine disorders (i.e., diabetes, thyroid diseases, etc.) and metabolic syndromes [3]. In addition, these hormones are widely abused in sports, either singly or simultaneously, as performance enhancers, often with little awareness of severe long-term side effects such as hypertension [4]. Thus detection and quantification of these hormones' levels in blood is crucial for both medical and sports-related applications.

The primary methods currently used to measure these hormone levels in blood are currently based on immunoassays and mass spectrometry as described in [5]. Immunoassays, such as enzyme-linked immunosorbent assays (ELISA), utilize specific antibodies to detect and quantify hormones based on their binding affinity [6]. Mass spectrometry, on the other hand, identifies hormones by measuring the mass-to-charge ratio of ionized molecules and provides high sensitivity and specificity [7]. However, both immunoassays and mass spectrometry have certain limitations. Immunoassays can suffer from cross-reactivity and are typically time-consuming due to the need for sample preparation and the use of antibodies. Mass spectrometry, while highly accurate, is often costly and requires advanced equipment, as well as skilled operators. Consequently, there is a strong need for developing innovative approaches for hormone detection.

Surface Enhanced Raman spectroscopy (SERS), a Raman variant technique well-suited for analyzing the sample's molecular composition and structural changes quantitatively [8], overcomes most of the listed disadvantages. SERS This technique offers significant advantages in that it is a label-free, non-destructive technique that can provide real-time, in situ measurements of hormone levels with high sensitivity and specificity [9,10]. Unlike immunoassays and mass spectrometry, SERS does not require antibodies or complex sample preparation [11,12]. It also allows for the detection of multiple biomarkers simultaneously, which is advantageous in analyzing complex biological samples [11]. Furthermore, SERS can be easily integrated into portable devices for on-site, point-of-care applications, making it an attractive alternative for rapid, cost-effective hormone level monitoring in clinical and field settings [5]. Although SERS holds significant potential for label-free detection of bio-molecular changes, interpreting the complex spectral information from biological samples presents numerous challenges [13,14]. For instance, its quantification procedure is cumbersome as SERS intensity is influenced by not only concentration but also instrumental effects such as spatial resolution and detector sensitivity [13]. Thus, quantifying chemicals, molecules, or substances using SERS spectral data is often difficult unless chemometric techniques like principal component analysis (PCA) and artificial neural networks (ANNs) are employed [15–17].

SERS combined with chemometrics has thus shown potential in biomedical fields in which hormone analysis such as estrogen [10,18], disease diagnosis such as cancer [19,20], and drug discovery [21] has been achieved. In addition, since this

technique provides quick and nondestructive measurements for biological sample examination [12], it is very promising for a variety of in-lab and on-site applications such as sport doping detection [22]. In detection of sport dopants, this technique overcomes most of the demerits faced by conventional techniques such as enzyme-linked immunoassay (ELISA) [2,23] and chromatographic methods [24]. Therefore, there is a great need to develop novel methods for singular or multiple detection of these dopants. In this work, SERS technique combined with ANN has been utilized for the detection of GH and TE hormones in blood from Sprague Dawley (SD) male rats.

## 2. Materials and methods

### 2.1. Animal handling and blood sampling

Twenty-four SD rats, fed and kept as described in [5], were divided into four groups: Non-injected rats, GH-injected rats, TE-injected rats, and rats injected with both GH and TE, with each group containing six rats. The rats received intramuscular injections of GH and TE either individually or simultaneously once a day, based on their weight. The rats underwent euthanasia following an overnight fasting period, achieved through intraperitoneal administration of 6% Phenobarbital on week and all efforts were made to minimize suffering. Subsequently, blood samples were obtained from each rat's orbital sinus at six time points: before injection, and at 30 minutes, 2 hours, 4 hours, 8 hours, and 24 hours after administration. This was to help in determining how long these hormones get elevated in blood before they go back to their normal concentrations. All experimental procedures complied with the FELASA guidelines on the use and care of laboratory animals. This study was ethically approved by Biosafety, Animal Use and Ethics Committee of Faculty of Veterinary Medicine, University of Nairobi under reference number "FVM BAUEC/2023/539"

### 2.2. SERS measurements

Thirty microliters of each of the blood samples obtained were mixed thoroughly with 150 µl of AgNPs (fabricated by laser ablation in liquids) with 34 nm diameter and spherical in shape as determined by SEM in [25]. Two microliters of the resulting mixture was then dropped onto aluminum-wrapped microscope glass slides (25.4 mm × 76.2 mm × 1.2 mm dimensions) and left to dry in air at room temperature for sixty (60) minutes. The SERS (Raman) measurements were performed on each sample as described in [5] using a portable Raman spectrometer (EZRaman-N Portable Analyzer System, Enwave Optronics, USA) with 150 mW laser excitation power.

### 2.3. ANN model development and validation methods

Two ANN models were trained and validated based on the six PC scores (explaining about 90% of the data variability) of simulate samples (blood with different concentrations of GH and TE ranging from 0.01 ng/ml to 60 ng/ml) obtained from the PCA process as described in [5]. To ensure robust training and validation, a k-fold cross-validation strategy was employed. The samples used were about 30 different concentrations (each having 30 spectra total to approximately 900 spectra) with 75% forming the training set and 25% making the test set. The input layer of these two models was PC scores and the output was predicted concentration. The hidden layer was made of 5 layers (arranged as 12:10:10:10:6 (for GH model) and 10: 8: 8: 8:6 (for TE model)) with rectifier (ReLU) activation function and resilient backpropagation (rprop+) algorithm. These model architectures were determined through experimentation, using empirical testing and optimization to balance accuracy and efficiency. The distinct architectures likely address the unique data complexity, variability, and performance requirements of each hormone model, improving the overall model performance and reproducibility. The six PC scores (capture approximately 90% of the data variability) were used rather than the full spectral data because these reduced the model's input dimensionality, leading to more efficient computation and potentially faster training without sacrificing significant information, as the six components explain the majority of the variance in the data.

The test for the accuracy of these models was done using the validation metrics such as Root Mean Squared Error (RMSE), and coefficient of determination ($R^2$) determined as shown in Equations (1) and (2) respectively.

$$RMSE = \sqrt{\left(\frac{1}{N}\sum_{j=1}^{N}(P_j - A_j)^2\right)} \tag{1}$$

$$R^2 = 1 - \frac{\sum_{j=1}^{n}(A_j - P_j)^2}{\sum_{j=1}^{n}(A_j - \overline{A}_j)^2} \tag{2}$$

In which $P_j$ represents the value of the predicted concentration, $A_j$ the known concentration value, N the total number of samples, $\overline{A}_j$ represents the average value of the known concentration. The detection limits, specifically the limit of detection (LOD) and limit of quantification (LOQ), were also evaluated. The LOD was determined using Equation (3) (for minimum LOD) and Equation (4) (for maximum LOD) [26].

$$LOD_{min} = 3.3 \times [(\sigma_b)^2(S_0)^{-2} + h_{0min}(\sigma_b)^2(S_0)^{-2} + h_{0min}(\sigma_{ycal})^2] \tag{3}$$

$$LOD_{max} = 3.3 \times [(\sigma_b)^2(S_0)^{-2} + h_{0max}(\sigma_b)^2 \times (S_0)^{-2} + h_{0max} \times (\sigma_{ycal})^2] \tag{4}$$

In which $\sigma_b$ represents the standard deviation of the response (SERS spectra of male rat's blood), $S_0$ represents the sensitivity of the calibration sample with minimum analyte concentration, $\sigma_{ycal}$ is the standard deviation of calibration errors (residues), and $h_{0min}$ and $h_{0max}$ are the minimum and maximum blank leverages. The value 3.3 is the expansion factors obtained assuming a 95% confidence interval. The minimum and maximum LOQ were calculated using the two respective equations (Eqs. 3 and 4) by replacing the value 3.3 with 10 [27,28]. Low values of minimum detection limits suggested good model performance. ANN was used in this study because it models complex, nonlinear relationships and adapts to variable data distributions without predefined forms when compared to other curve-fitting methods. It can also efficiently handle high-dimensional inputs, generalize well on large datasets, and are more robust to noise, while curve-fitting methods may struggle with these challenges and require simpler model assumption. On ensuring that the models were accurate, the models were then used to predict the concentrations of these hormones (GH and TE) in blood from rats not injected and those injected with GH only, TE only, and both GH and TE. All these was done in MATLAB R2021a environment (version 9.10.0.1602886, The MathWorks Inc., Natick, USA).

## 3. Results and discussions

### 3.1. Qualitative analysis of blood from injected and non-injected rats

To identify the Raman intensity bands associated with GH, TE and both hormones (GH and TE) in blood of SD rats, the normalized and averaged SERS spectra (derived from thirty spectra per group) for blood from SD rats injected with GH only, TE only, both hormones and non-injected rats is displayed in Fig 1.

The SERS spectra of blood from non-injected and injected rats, as shown in Fig 1, appear nearly identical, with only slight variations in band intensity. These minor differences suggest subtle yet distinct biochemical changes resulting from hormone injections [29]. Rather than causing significant alterations in molecular composition, GH and TE injections likely influence specific biochemical pathways, leading to these variations. Such changes may involve slight modifications in protein conformation, lipid interactions, or hormone-related molecular rearrangements, which remain detectable through SERS despite the overall spectral similarity. The prominent bands noted were those centered around wavenumbers 658, 798, 878, 914, 932, 1064, 1190, 1354, 1410, and 1658 cm$^{-1}$. These bands were found to be related to those obtained from blood, and blood mixed with various concentrations of these hormones [5]. The bands centered around 600–1064 cm$^{-1}$

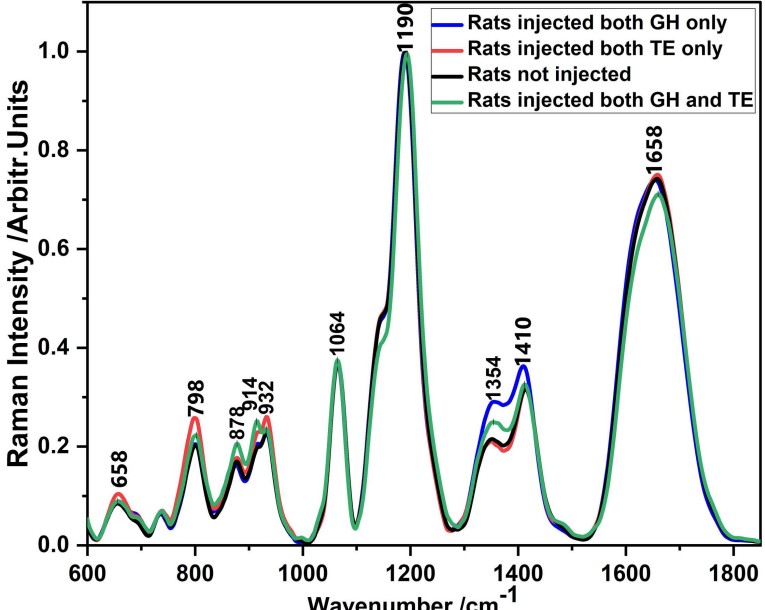

**Fig 1. SERS spectra for blood from rats injected with GH only, TE only, both GH and TE hormones, and non-injected rats.**

were assigned to C-C stretching [10,30–32], 1190 cm⁻¹ assigned to C-O Stretch and COH bending [33], 1354 cm⁻¹ assigned to Stretch (COO-) and C-H bend [30], and 1658 cm⁻¹ assigned to C-O/C=O Stretching [8,10]. The SERS bands with the highest intensity were those centered at around 658, 798, 932, and 1658 cm⁻¹ for blood from rats injected with TE only; 878 and 914 cm⁻¹ for blood from rats injected with both hormones; and 1354 and 1410 cm⁻¹ for blood from rats injected with GH only. This analysis underscores the sensitivity of SERS in detecting hormonal influences on blood composition, providing a valuable tool for studying the biochemical effects of hormone treatments.

PCA was used to further investigate if there were differences in spectral patterns of the blood from non-injected rats and those injected with each of the two hormones. Fig 2 shows the PCA scores and loadings plots for blood from GH-injected rats, TE-injected rats, and non-injected rats. The displayed score plots show clear segregation of the SERS spectral data sets for blood from GH-injected rats, TE-injected rats, and non-injected rats thus supporting the idea that the SERS spectroscopic technique is a potentially sensitive alternative hormonal assaying method (see Fig 2a). The three PCs explained 81.5% of the variability of the data, where PC1, PC2, and PC3 accounted for 43.2%, 23.2%, and 15.1% respectively. The SERS bands that brought about this segregation were determined using PCA loadings plot based on the PCA score segregation (see Fig 2b (i–iii)). These bands are those centered at wavenumbers 684, 1042, 1378, and 1596 cm⁻¹ for GH-injected rats (see Fig 2b (iv)); 798, 912, and 1690 cm⁻¹ for TE-injected rats (see Fig 2b (v)); and 1206, 1456 and 1722 cm⁻¹ for non-injected rats (see Fig 2b (vi)). Some of these bands matched exactly with some bands identified as concentration sensitive bands using simulate samples as shown by [5].

To check whether there was spectral variation for different times before and after hormone injection, the average SERS spectra for each of the groups was plotted for different times - before injection and 0.5, 2, 4, 8, and 24 hours after injection as shown in Fig 3. From Fig 3, it is seen that the spectral profiles are very similar in each of the hormones for different times before and after injection. This likely means that some molecular constituents or metabolic pathways have relatively stable spectral signatures against the background of dynamic metabolic processes in response to hormone injection. Although the SERS spectral profiles obtained were generally very similar, subtle variations in SERS band intensities at specific wavenumbers of different times of the day and hormone groups are apparent. Noticeably, bands centered at 1350

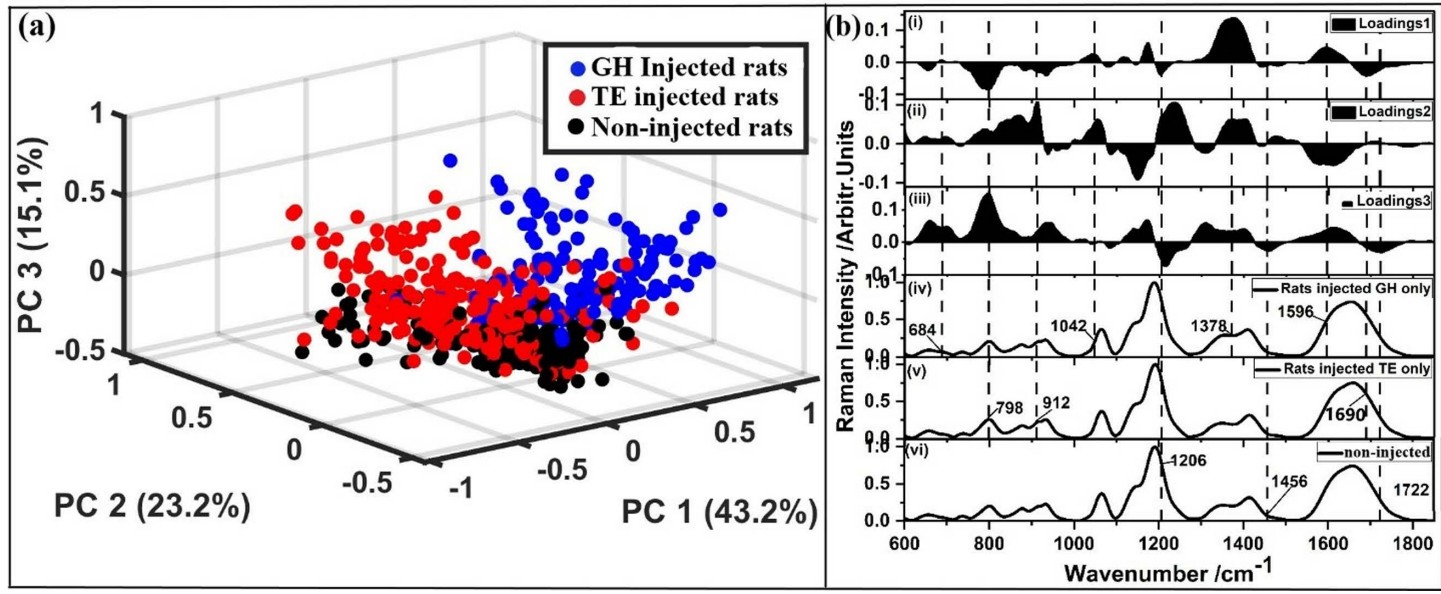

**Fig 2. Three-dimensional PCA (a) score plots, (b) loadings plot for (i) PC1, (ii) PC2, and (iii) PC3 loadings plot, and averaged SERS spectra of blood from (iv) rats injected with GH only, (v) rats injected with TE only and (vi) non-injected rats.** The explained variances are indicated in percentages and were 43.2%, 23.2%, and 15.1% for PC1, PC2, and PC3 respectively.

and 1410 cm⁻¹ show apparent differences related to non-injected rats, GH-injected rats, and TE-injected rats. Also, the band at 798 cm⁻¹ has some variation, specifically with the GH injection. This may be an indication of the fact that these variations in metabolic responses of the hormone-specific molecular rearrangements within the tissue microenvironment mean an elevation of these hormones after an injection. To validate the observed differences, statistical analysis, including descriptive statistics and one-way ANOVA, was conducted. The results yielded standard deviations of 0.05362 (for 1350 cm⁻¹), 0.04208 (for 1410 cm⁻¹), and 0.06452 (for 798 cm⁻¹), with variance values ranging between 0.003 and 0.004 for the three bands. These standard deviation and variance values indicate significant molecular response variations, potentially supporting the hypothesis that GH and TE injections induce molecular rearrangements detectable through spectroscopy.

To determine further if there was spectral variation for different times before and after hormone injection, PCA was performed for each group for each at different times (before injection and 0.5, 2, 4, 8, and 24 hours after injection) (see Fig 4). As displayed in Fig 4, the first three PCs explained 82% (GH injected rats), 81.1% (TE injected rats), 76.8% (GH+TE injected rats), and 69.5% (non-injected rats) of the variance suggesting that the spectral profiles of blood obtained at different times were different. GH-injected rats had the highest variance, closely followed by TE-injected rats, while variance reduced in rats injected simultaneously with both hormones when compared to those injected singly. Non-injected rats exhibited the least variance, indicating more consistent spectral profiles.

The loadings plot displayed in Fig 5 shows that the bands responsible for the segregation of the data sets in the score plots were centered at around 1378 cm⁻¹ (for all groups); 658 cm⁻¹, 1614 cm⁻¹ (for GH injected rats), 798 cm⁻¹ (separately GH and TE injected rats); 786 cm⁻¹ (non-injected rats and rats injected with both GH and TE); 914 and 1240 cm⁻¹ (TE injected rats and rats injected both GH and TE); 876 and 1636 cm-1 (rats injected both GH and TE). These bands could be used as Raman biomarker bands for periodic concentration (level) changes in blood for the respective hormones. These results also point to the great power of the SERS method in detecting subtle respective hormone level changes in blood.

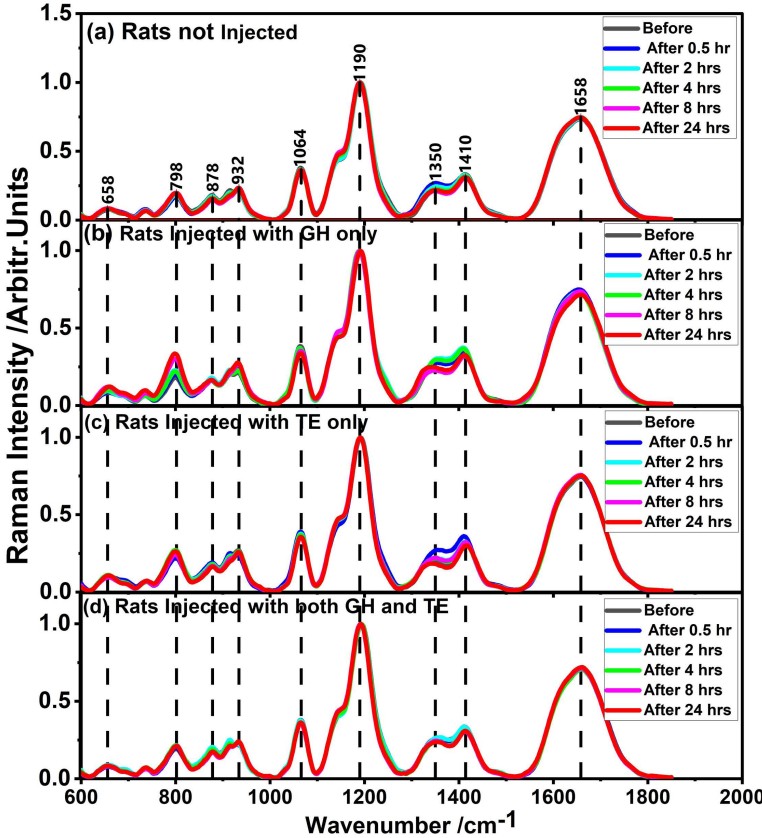

**Fig 3. SERS spectra of different for blood from (a) non-injected rats, (b) rats injected with GH only, (c) rats injected with TE only, and (d) rats injected with both GH and TE hormones.** The blood was drawn at different times before and after injection.

### 3.2. Quantitative analysis of GH and TE hormones

Blood is a complex matrix composed of a myriad number of Raman active biological molecules with isoenergetic vibrational bands. In order to quantify the levels of the hormones of interest in this work, GH and TE, in blood multivariate analytical methods are necessary. In this work, artificial neural network (ANN) were employed in performing concentration determination in the investigated blood samples. Two ANN models (one for TE and the other for GH) were trained, validated and used to predict each hormone's concentration in blood. Table 1 displays validation results obtained from these developed ANN models.

The success of the model is normally assessed by the high (low) values of the coefficient of determination, $R^2$ (low values of the root-mean square error, RMSE). Indeed, as seen from Table 1, the RMSE values were low and the $R^2$ values were close to one which was an indication that these models were suitable for performing level predictions of these hormones in blood using the SERS data sets. Further performance evaluation of the models were assessed using the regression plots of predicted concentration versus the known (see Fig 6). In all models, data were close to the regression lines revealing that the predicted concentration of both the training and testing set was in acceptable good agreement with the experimental (actual) concentration. This confirmed the excellent performance of the models.

Using Equations 3 and 4 evaluated were also the detection limits, i.e., LOD and LOQ of the SERS method used in our work (see Table 2). The minimum LOD deduced were 0.3222 ng/ml (for GH) and 0.1851 ng/ml (for TE) while the minimum LOQ determined were 0.9766 ng/ml (for GH) and 0.5611 ng/ml (for TE). This revealed that the LOD and LOQ

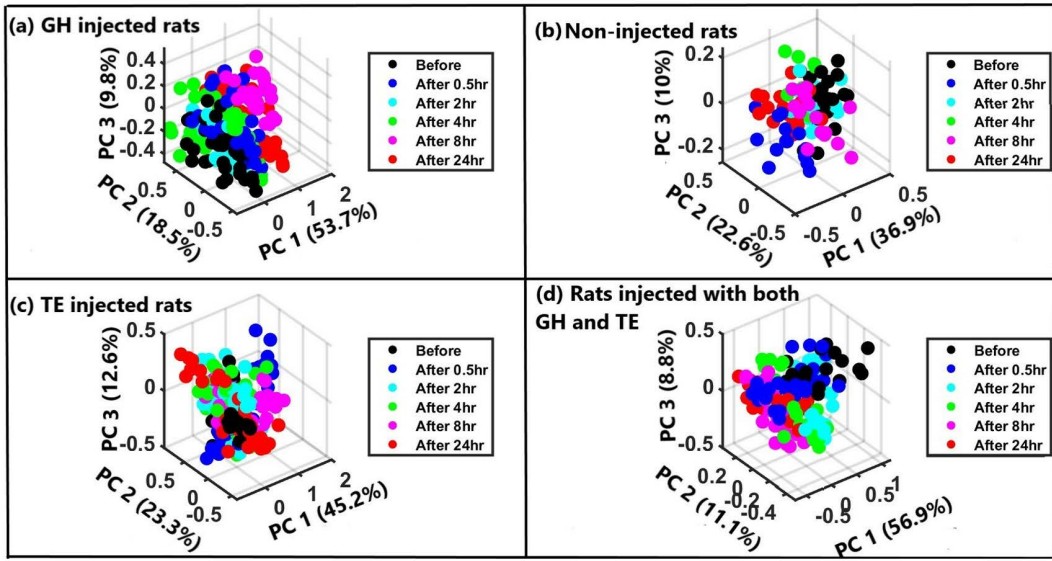

**Fig 4. Three-dimensional PCA score plots for (a) rats injected with GH only, (b) Normal rats, (c) rats injected with TE only, and (d) rats injected simultaneously with GH and TE based on different times before and after injection.**

values obtained for TE enanthate were notably in range with those reported by other analytical methods, as evidenced by the comparative data presented in Table 2. For instance, from previous studies done on detection of TE using dried blood samples, the LOD obtained ranged from 0.05 to 0.2 ng/ml when using NanoLC-HRMS/MS technique depending on the type of TE injected intramuscularly [34]. Similarly, a LOD of 0.2 ng/ml and a LOQ of 0.5 ng/ml for TE enanthate was obtained when using dried blood samples extraction coupled with TurboFlow technology and LC-HRMS by means of the DBSA-TLXHRMS system [35]. This implies that the SERS method combined with ANN models offers superior sensitivity in detecting and quantifying TE enanthate, thereby enhancing the precision and reliability of measurements in related research and clinical applications. When using GH ELISA assays the LOD and LOQ obtained were 0.04 and 0.065 ng/ml respectively [28,36], while using isotope dilution mass spectrometry (IDMS) the values were 0.5 ng/ml (LOD) and 0.7 ng/ml (LOQ) [37]. Although the GH minimum detection limits obtained in this study are seen to be higher when compared to ELISA and lower when compared to IDMS (Table 2) these values (LOD and LOQ) may still be acceptable since they are very low when compared with the range of GH concentrations typically encountered in the rat blood samples (3–25 ng/ml). Upon ascertaining that these models were accurate and had acceptable detection limits, they were used to detect the levels GH and TE hormones in blood of rats injected singly and simultaneously with the two hormones in comparison to those not injected. This is because elevation in concentrations of these hormones in blood above the normal or known concentrations might suggest doping which is often done by athletes to increase their sport activity [38].

The average values of the predicted concentration levels of the respective hormones were determined before and some hours after injection (0.5, 2, 4, 8, and 24 hours) in each of the groups (injected and non-injected). Fig 7 displays GH and TE hormone levels (on average) in blood taken from GH with/and TE injected and non-injected SD rats throughout the 24 hours of study.

The administration of GH led to a rapid elevation in GH levels within half an hour, reaching a peak concentration of 44 ng/ml, indicating the prompt bioavailability and uptake of exogenous GH. This elevation in GH levels was sustained for a brief period before gradually declining, although remaining elevated compared to baseline levels (Fig 6a). The return of GH levels to normal after 24 hours suggests the transient nature of exogenous GH effects, with hormonal homeostasis

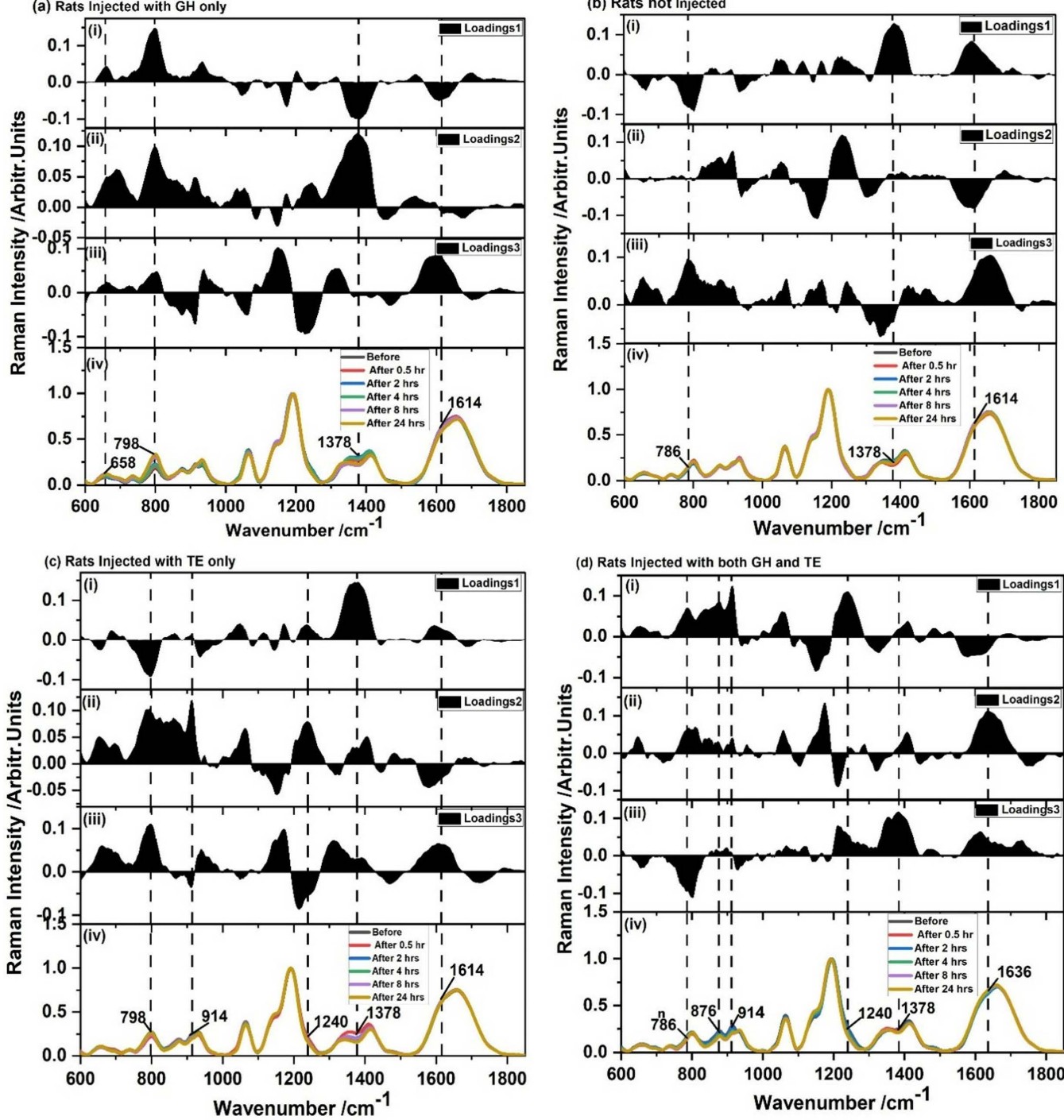

**Fig 5. PCA loading plots and SERS Spectra of blood from rats injected with (a) GH only, (b) non-injected rats, (c) rats injected with TE only, and (d) rats injected simultaneously with GH and TE hormones drawn at different times before and after injection.**

**Table 1. Validation metrics of the ANN models constructed, trained and validated.**

| Model | RMSE (ng/ml) | | $R^2$ | |
|---|---|---|---|---|
| | Training | Validation | Training | Validation |
| Growth hormone | 0.54489 | 0.64364 | 0.9135 | 0.8771 |
| Testosterone | 0.23870 | 0.42386 | 0.9804 | 0.9331 |

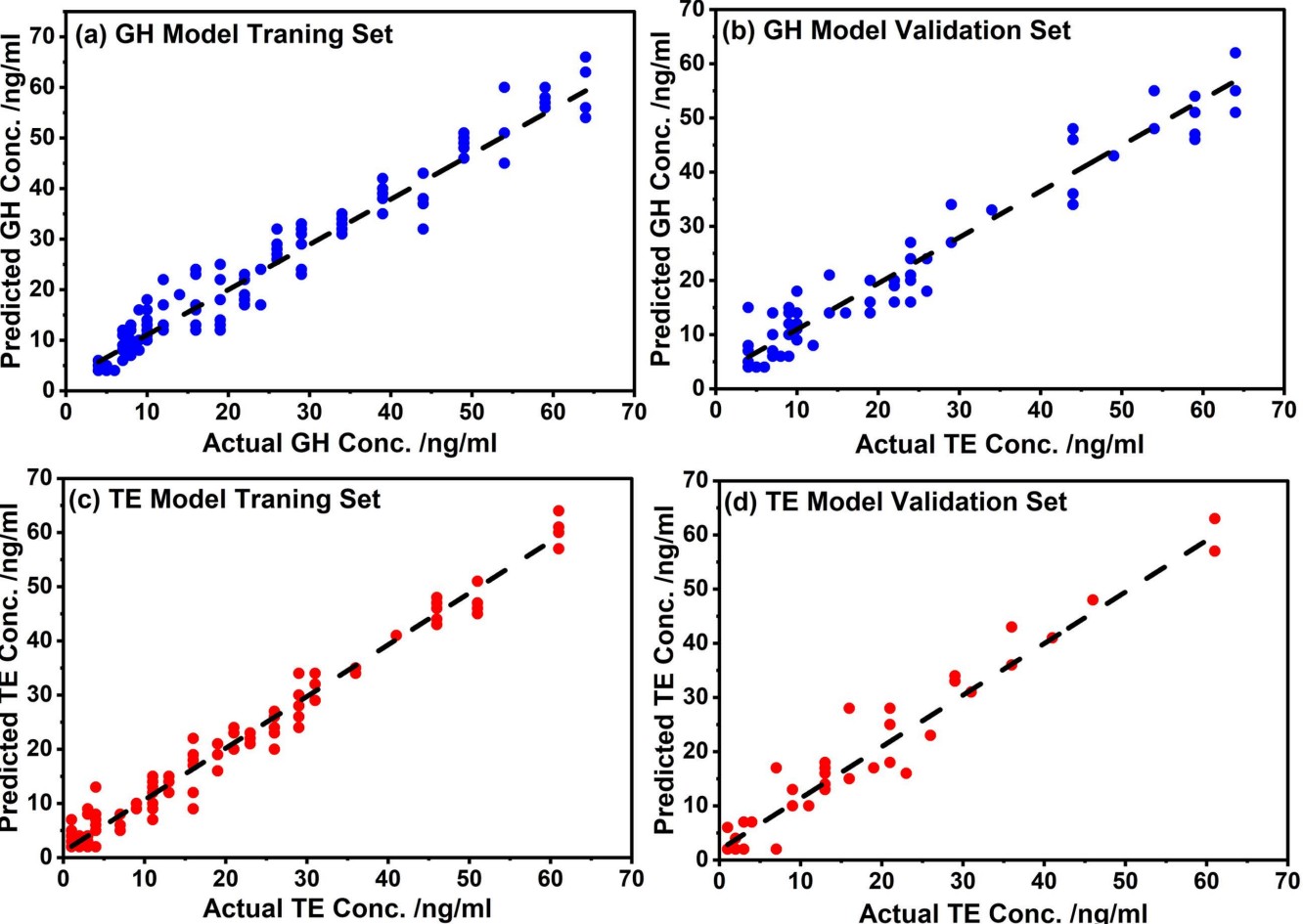

**Fig 6. ANN models' regression curves showing predicted concentration against actual concentration.**

being restored over time. Similarly, administration of TE resulted in a progressive increase in TE levels over time, peaking between half an hour and eight hours post-injection (Fig 6b), consistent with the expected pharmacokinetic profile of TE. Interestingly, concurrent administration of TE and GH did not significantly alter the temporal pattern of TE elevation compared to TE alone, indicating independent mechanisms of action for these hormones [39]. Importantly, TE levels returned to baseline after 24 hours across all treatment groups, underscoring the transient nature of TE effects and the efficient metabolism and elimination of exogenous TE. The observed changes in hormone levels highlight the dynamic interplay between exogenously administered hormones and endogenous hormonal regulation mechanisms. The rapid elevation and subsequent decline in GH levels reflect the pulsatile nature of GH secretion and the regulatory feedback mechanisms

**Table 2. The comparison between the detection limits for the ANN models created in this work with those published in the literature.**

| Hormone | Limit of detection (LOD) (ng/ml) | | Limit of quantification (LOQ) (ng/ml) | | Reported LOD in literature (ng/ml) | Reported LOQ in literature (ng/ml) | References |
|---|---|---|---|---|---|---|---|
| | $LOD_{min}$ | $LOD_{max}$ | $LOQ_{min}$ | $LOQ_{max}$ | | | |
| Growth hormone | 0.3222 | 42.33 | 0.9766 | 128.28 | 0.04 (ELISA) and 0.5 (IDMS) | 0.065 (ELISA) and 0.7 (IDMS) | [28,36,37] |
| Testosterone | 0.1851 | 18.57 | 0.5611 | 56.28 | 0.2 (LC-HRMS) and 0.05–0.2 (NanoLC-HRMS/MS) | 0.5 (LC-HRMS) | [34,35] |

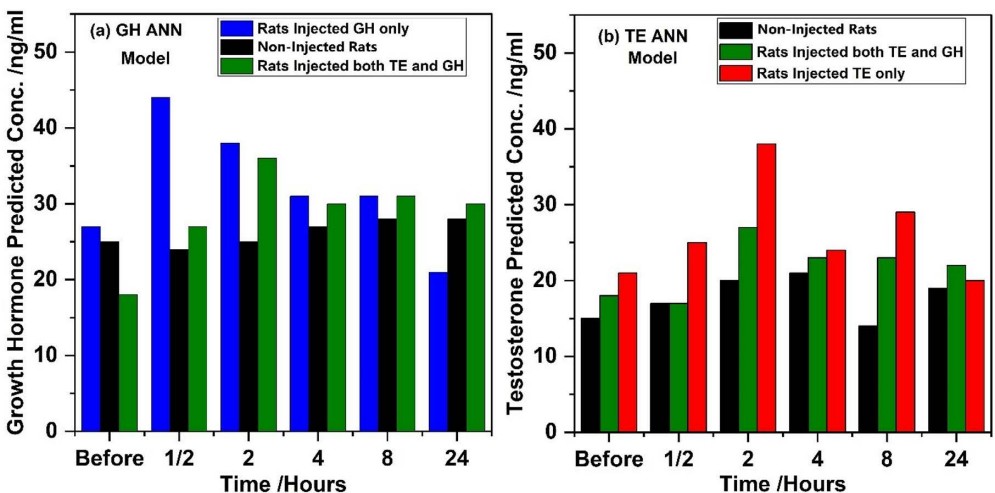

**Fig 7. Predicted concentrations using the (a) GH and (b) TE ANN models.**

that maintain hormonal balance [38]. Similarly, the temporal profile of TE levels reflects its pharmacokinetic properties, including absorption, distribution, metabolism, and elimination processes.

To validate the data obtained, ELISA was used and the results are displayed in Fig 8. Here ELISA standard curves of each hormone was plotted as a graph of optical density against the known concentration as in Fig 8a for GH and Fig 8b for TE. A best-fit curve, i.e., a Boltzmann function (for GH) and exponential decay function (for TE), was then applied to these points to establish a relationship between concentration and optical density. This fit allowed for prediction of the concentrations of GH and TE hormones in unknown blood samples by comparing their optical density to the standard curve (see Fig 8c and 8d).

As seen from the standard curves, the data showed an excellent fit, with the coefficient of determination values being nearly one, i.e., 0.99946 for GH (Fig 8a) and 0.99733 for TE (Fig 8b). The predicted concentration by ELISA, was noted to correlate with those concentrations obtained by the ANN models. Initially, GH concentrations in both non-injected rats and rats injected with hormones were within normal ranges (3 ng/ml), indicating baseline physiological levels. However, following GH injection, GH concentrations rapidly increased, reaching peak levels (26 ng/ml) within two hours post-injection, before returning to baseline levels by four hours. This temporal profile suggests a transient elevation in GH concentrations, with exogenously administered GH being detectable in the bloodstream for up to two hours post-injection. Beyond this timeframe, GH concentrations reverted to normal levels, underscoring the limited duration of exogenous GH detection using the ELISA kit employed in our study. Conversely, TE levels at baseline were also within normal ranges in both non-injected rats and rats injected with hormones (2 ng/ml). Following TE injection, TE concentrations gradually increased, peaking at two hours for rats injected with TE alone and at eight hours for rats injected simultaneously with GH and TE.

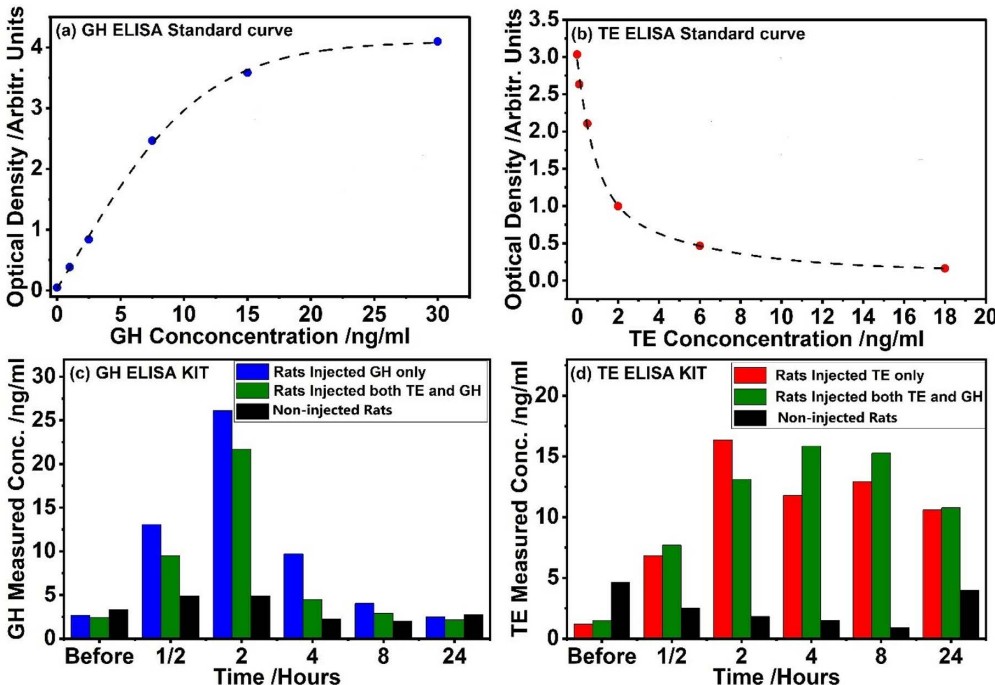

**Fig 8. ELISA results for non-injected rats and rats injected singly and simultaneously with GH and TE hormones.**

This sustained elevation in TE concentrations suggests a prolonged presence of exogenously administered TE in the bloodstream, with detectable levels lasting up to eight hours post-injection. Importantly, these findings are consistent with results obtained from SERS with ANN models, providing additional validation of the ELISA kit results.

The accuracy and dependability of various analytical methods for measuring hormone concentrations after exogenous injection can be better understood by comparing the results of ELISA and SERS with ANN. In this study, significant similarities as well as differences between the two approaches' temporal profiles of GH and TE concentrations were noted. As demonstrated by the findings of the ELISA and SERS with ANN, the injection of GH triggered a rapid and sustained rise in GH levels within the first two hours after the injection. The GH concentration that peaked between 30 minutes and 2 hours after injection and then gradually decreased is consistent with the pharmacokinetic profile of GH that is given exogenously. This consistency between SERS with ANN and ELISA results underscores the reliability of both techniques in capturing the dynamic changes in GH concentrations over time. Similarly, the administration of TE led to a progressive increase in TE levels over time, peaking between half an hour and eight hours post-injection, as observed with both SERS with ANN and ELISA methods. The sustained elevation in TE concentrations and the prolonged detection window of up to eight hours post-injection are consistent across both analytical techniques, reflecting the pharmacokinetic properties of exogenously administered TE. Additionally, both SERS with ANN and ELISA results showed that simultaneous injection of TE and GH did not significantly change the temporal patterns of TE concentrations compared to TE alone. It indicates that TE and GH have separate modes of action, with neither hormone significantly interfering with the other's ability to affect hormone concentrations. The validity of SERS with ANN in capturing the temporal changes of GH and TE concentrations is therefore demonstrated by the consistency of the hormone quantification results between SERS with ANN and ELISA technique. This demonstrates that the combination of SERS with ANN provides a label-free, non-destructive approach for multiplexed hormone measurement, with possible advantages in terms of sample throughput and adaptability.

## 4. Conclusion

This work has demonstrated the ability of SERS together with ANN models in quantifying the GH and TE hormone levels in blood of male SD rats. Here, two ANN models were developed based on six PC scores obtained from the PCA of different concentrations of each of the hormones in blood. A test for the model's accuracy was done using $R^2$ and RMSE values that were seen to be greater than 87.71% and less than 0.6436 respectively implying the models were accurate. The minimum limit of detection (LOD) deduced were 0.3222 ng/ml (for GH) and 0.1851 ng/ml (for TE) while the minimum limit of quantification (LOQ) determined were 0.9766 ng/ml (for GH) and 0.5611 ng/ml (for TE). This detection limits were very low and thus acceptable. Using the calibrated ANN models in determining the concentration hormone level in blood of rats, it was noted that hormone injected rats' respective hormone levels elevated for some time and declined later when compared to those of non-injected rats. This implied that exogenous injection of sport dopants can be detected and quantified using SERS technique combined with ANN models. This could bring about the customization of a SERS system that utilizes SERS data and inbuilt ANN models to detect GH and TE levels in blood in less than a minute. These findings widen the potential use of SERS in sports science, clinical diagnostics, and biomedical research.

## Author contributions

**Conceptualization:** Zephania Birech, Peter W. Mwangi.

**Data curation:** Annah M. Ondieki, Moses Juma, Boniface M. Chege.

**Formal analysis:** Annah M. Ondieki, Moses Juma.

**Funding acquisition:** Zephania Birech, Kenneth A. Kaduki.

**Investigation:** Annah M. Ondieki, Boniface M. Chege.

**Methodology:** Annah M. Ondieki, Peter W. Mwangi, Moses Juma, Boniface M. Chege.

**Project administration:** Zephania Birech, Kenneth A. Kaduki, Peter W. Mwangi.

**Resources:** Zephania Birech, Kenneth A. Kaduki, Peter W. Mwangi.

**Software:** Annah M. Ondieki.

**Supervision:** Zephania Birech, Kenneth A. Kaduki, Peter W. Mwangi.

**Validation:** Annah M. Ondieki.

**Visualization:** Annah M. Ondieki.

**Writing – original draft:** Annah M. Ondieki.

**Writing – review & editing:** Annah M. Ondieki, Zephania Birech, Kenneth A. Kaduki, Peter W. Mwangi.

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
