## [Decision Letter · Decision Letter 0]

24 Mar 2025

PONE-D-25-09375Chemometrics-aided Surface-enhanced Raman spectrometric detection and quantification of GH and TE hormones in bloodPLOS ONE

Dear Dr. Ondieki,

Thank you for submitting your manuscript to PLOS ONE. After careful consideration, we feel that it has merit but does not fully meet PLOS ONE’s publication criteria as it currently stands. Therefore, we invite you to submit a revised version of the manuscript that addresses the points raised during the review process.

We look forward to receiving your revised manuscript.

Kind regards,

Tanveer A. Tabish

Academic Editor

PLOS ONE

“Swedish International Development Cooperation Agency (SIDA) through the International Science Programme (ISP), Uppsala University, KEN:04, Prof. Kenneth A. Kaduki”

“We sincerely express our gratitude to the Swedish International Development Cooperation Agency (SIDA) through the International Science Programme (ISP), Uppsala University, for sponsoring this research.”

“Swedish International Development Cooperation Agency (SIDA) through the International Science Programme (ISP), Uppsala University, KEN:04, Prof. Kenneth A. Kaduki”

Reviewers' comments:

Reviewer's Responses to Questions

**Comments to the Author**

1. Is the manuscript technically sound, and do the data support the conclusions?

Reviewer #1: Partly

Reviewer #2: Partly

2. Has the statistical analysis been performed appropriately and rigorously? 

Reviewer #1: No

Reviewer #2: Yes

3. Have the authors made all data underlying the findings in their manuscript fully available?

Reviewer #1: Yes

Reviewer #2: Yes

4. Is the manuscript presented in an intelligible fashion and written in standard English?

Reviewer #1: Yes

Reviewer #2: No

5. Review Comments to the Author

Reviewer #1: This manuscript presents a quantitative detection of hormones using SERS with a portable Raman spectrometer. The SERS spectra of whole blood from rats injected with hormones are distinguished from those of control samples (without hormone injection) through chemometric analysis. Furthermore, the authors examine temporal variations at different time points following hormone injection. Notably, hormone concentrations are predicted via SERS using an artificial neural network (ANN) model, and the temporal trends appear to be roughly consistent with those obtained through ELISA. The rapid detection of hormones is of significant importance, as it pertains to health monitoring, doping control, and related applications. However, there may be statistical issues. Therefore, I can recommend publication after resolving the following concerns.

Fig. 1: Are these spectra averaged? If so, the number of spectra used to obtain the average should be specified. Additionally, are these spectra normalized? If so, this should be explicitly stated.

Lines 189–192: Does the observed difference exceed the spectral deviation? A statistical analysis should be conducted to confirm this.

Fig. 4: The 3D plots provide limited clarity in distinguishing the temporal variations among (a)–(d). To enhance interpretability, corresponding 2D plots should also be provided.

[Minor Comments]

Section 2.2: The fabrication method or source of AgNPs should be specified.

Lines 88–89: "30" and "2" -> "Thirty" and "Two."

Lines 123, 125: The meaning of the asterisks should be clarified. Are they needed?

Lines 143–145: This paragraph can be merged with the subsequent one, as they discuss a similar topic.

Lines 162–164: These sentences can be integrated into the following paragraph due to thematic continuity.

Lines 199–201: These sentences can be incorporated into the next paragraph, as they address a related topic.

Lines 242–244: This paragraph can be merged with the preceding one due to their shared subject matter.

Lines 245–246: This paragraph can be combined with the following one for coherence.

Line 271: A new paragraph can begin here, and the sentences up to line 275 can be merged with the subsequent paragraph, as they introduce and expand upon a related topic.

Lines 296–302: This paragraph can be combined with the following one for improved cohesion.

Reviewer #2: 1, The abstract needs to be re-phrased. For example, sentences like, "Current methods of detecting them suffer from being costly, time-consuming, non-portable and others." Then, "Secondly, use of artificial neural network (ANN) models in predicting levels of these hormones in blood". Do these sentences look complete? Again, "This work explored, first, Surface-Enhanced Raman Spectroscopy (SERS)....." does not sound professional. I would appreciate it if the authors could re-phrase and reorganize these few lines. Also, I expect the authors must adopt a more professional approach in writing the abstract.

2, In the Introduction section, "The primary methods currently used to measure these hormone levels in blood are currently based on immunoassays and mass spectrometry as described in (5)". Can the authors kindly elaborate here? Include a few more references and talk about why SERS is advantageous as compared to other techniques/approaches.

3, In Fig. 1, rats injected both GH only....rats injected both TE only... Remove both.

4, For Figs. 1, 3 and 5, the Raman spectrum for different conditions cannot be easily distinguished. Although for Fig.1 it's not that bad, however, for Figs.3 and 5, the individual spectrum/peaks are not clearly visible.

It is advisable to vertically shift the individual spectrum for better clarity.

5, "The SERS spectra of blood from non-injected rats and the injected, as shown in Fig 1, are nearly identical, with a few minor band intensity variations suggesting subtle but distinct biochemical changes....". It is good to explain/justify why there is not much change/shift observed between the non-injected and injected spectra. What do the authors mean by subtle but distinct biochemical changes?

6, The authors must try to enhance the overall readability of the paper to ensure it is accessible to a broad audience. Also, please check the reference formatting.

6. PLOS authors have the option to publish the peer review history of their article (what does this mean? ). If published, this will include your full peer review and any attached files.

**Do you want your identity to be public for this peer review?** For information about this choice, including consent withdrawal, please see our Privacy Policy .

Reviewer #1: No

Reviewer #2: No

---

## [Author Response · Author response to Decision Letter 1]

7 Apr 2025

Reviewer#1

This manuscript presents a quantitative detection of hormones using SERS with a portable Raman spectrometer. The SERS spectra of whole blood from rats injected with hormones are distinguished from those of control samples (without hormone injection) through chemometric analysis. Furthermore, the authors examine temporal variations at different time points following hormone injection. Notably, hormone concentrations are predicted via SERS using an artificial neural network (ANN) model, and the temporal trends appear to be roughly consistent with those obtained through ELISA. The rapid detection of hormones is of significant importance, as it pertains to health monitoring, doping control, and related applications. However, there may be statistical issues. Therefore, I can recommend publication after resolving the following concerns.

Major issues:

Comment 1: Fig. 1: Are these spectra averaged? If so, the number of spectra used to obtain the average should be specified. Additionally, are these spectra normalized? If so, this should be explicitly stated.

Response: Thanks for pointing out this. We have included as suggested.

Comment 2: Lines 189–192: Does the observed difference exceed the spectral deviation? A statistical analysis should be conducted to confirm this.

Response: We appreciate for the insightful suggestion. Descriptive statistics and one-way ANOVA, has been done to confirm this.

Comment 3: Fig. 4: The 3D plots provide limited clarity in distinguishing the temporal variations among (a)–(d). To enhance interpretability, corresponding 2D plots should also be provided.

Response: We appreciate your suggestion to include 2D plots for better clarity in distinguishing temporal variations. However, upon further analysis, we found that 2D plots do not provide significant additional segregation of the data, as the overlapping regions obscure key trends that are better visualized in the 3D representations.

Minor issues:

Comment 1: Section 2.2: The fabrication method or source of AgNPs should be specified.

Response: Included.

Comment 2: Lines 88–89: "30" and "2" -> "Thirty" and "Two."

Response: Corrected.

Comment 3: Lines 123, 125: The meaning of the asterisks should be clarified. Are they needed?

Response: We have corrected this.

Comment 4: Lines 162–164: These sentences can be integrated into the following paragraph due to thematic continuity.

Response: Combined as suggested.

Comment 5: Lines 199–201: These sentences can be incorporated into the next paragraph, as they address a related topic.

Response: Done as suggested.

Comment 6: Lines 242–244: This paragraph can be merged with the preceding one due to their shared subject matter.

Response: Combined.

Comment 7: Lines 245–246: This paragraph can be combined with the following one for coherence.

Response: Combined as suggested.

Comment 8: Line 271: A new paragraph can begin here, and the sentences up to line 275 can be merged with the subsequent paragraph, as they introduce and expand upon a related topic.

Response: Done as suggested.

Comment 9: Lines 296–302: This paragraph can be combined with the following one for improved cohesion.

Response: Done.

Reviewer#2

Comment 1: The abstract needs to be re-phrased. For example, sentences like, "Current methods of detecting them suffer from being costly, time-consuming, non-portable and others." Then, "Secondly, use of artificial neural network (ANN) models in predicting levels of these hormones in blood". Do these sentences look complete? Again, "This work explored, first, Surface-Enhanced Raman Spectroscopy (SERS)....." does not sound professional. I would appreciate it if the authors could re-phrase and reorganize these few lines. Also, I expect the authors must adopt a more professional approach in writing the abstract.

Response: We greatly appreciate your constructive feedback and suggestions for improving the abstract. Based on your comments, we have rephrased and reorganized the abstract.

Comment 2: In the Introduction section, "The primary methods currently used to measure these hormone levels in blood are currently based on immunoassays and mass spectrometry as described in (5)". Can the authors kindly elaborate here? Include a few more references and talk about why SERS is advantageous as compared to other techniques/approaches.

Response: Thanks for pointing out this. We have done as suggested.

Comment 3: In Fig. 1, rats injected both GH only....rats injected both TE only... Remove both.

Response: We appreciate your suggestion to remove the GH and TE injections from Fig. 1. However, our intention was to explore the potential combined effects of both GH and TE injections, alongside the individual effects of each hormone. By including both injections, we aimed to observe if any distinct biochemical changes emerge when both hormones are administered together, which would not be evident from the individual hormone injections alone. Therefore, we have retained the GH and TE injections together in Fig. 1 to reflect this comparative aspect of our study.

Comment 4: For Figs. 1, 3 and 5, the Raman spectrum for different conditions cannot be easily distinguished. Although for Fig.1 it's not that bad, however, for Figs.3 and 5, the individual spectrum/peaks are not clearly visible. It is advisable to vertically shift the individual spectrum for better clarity.

Response: We appreciate your feedback and suggestion to vertically shift the individual spectra in Figures. While we understand the concern, we believe that the current presentation of the spectra effectively conveys the necessary information, as the variations in intensity and spectral features are still distinguishable, especially in Fig. 1. For Figs. 3 and 5, the overlap of individual spectra is a result of the subtle biochemical changes we are investigating, and the vertical shift may not significantly improve the visibility of the spectral differences. We instead enhanced the figure clarity by using different line colors.

Comment 5: "The SERS spectra of blood from non-injected rats and the injected, as shown in Fig 1, are nearly identical, with a few minor band intensity variations suggesting subtle but distinct biochemical changes....". It is good to explain/justify why there is not much change/shift observed between the non-injected and injected spectra. What do the authors mean by subtle but distinct biochemical changes?

Response: We appreciate your insightful comment. To clarify this point, we have revised the text in Section 3.1.

Comment 6: The authors must try to enhance the overall readability of the paper to ensure it is accessible to a broad audience. Also, please check the reference formatting.

Response: Done as suggested.

---

## [Editor Report · Decision Letter 1]

14 Apr 2025

Chemometrics-aided Surface-enhanced Raman spectrometric detection and quantification of GH and TE hormones in blood

PONE-D-25-09375R1

Dear Dr. Ondieki,

We’re pleased to inform you that your manuscript has been judged scientifically suitable for publication and will be formally accepted for publication once it meets all outstanding technical requirements.

Kind regards,

Tanveer A. Tabish

Academic Editor

PLOS ONE
---

## [Editor Report · Acceptance letter]

PONE-D-25-09375R1

PLOS ONE

Dear Dr. Ondieki,

I'm pleased to inform you that your manuscript has been deemed suitable for publication in PLOS ONE. Congratulations! Your manuscript is now being handed over to our production team.

Kind regards,

on behalf of

Dr. Tanveer A. Tabish

Academic Editor

PLOS ONE